# A New Semi-Analytical MC Model for Oceanic LIDAR Inelastic Signals

**Su Chen** [1,2,3], **Peng Chen** [2,3,4], **Lei Ding** [1,3,*] **and Delu Pan** [2,4,5]

1  Key Laboratory of Infrared System Detection and Imaging Technologies, Shanghai Institute of Technical Physics, Chinese Academy of Sciences, Shanghai 200083, China
2  State Key Laboratory of Satellite Ocean Environment Dynamics, Second Institute of Oceanography, Ministry of Natural Resources, Hangzhou 310012, China
3  School of Electronic, Electrical and Communication Engineering, University of Chinese Academy of Sciences, Beijing 100049, China
4  Donghai Laboratory, Zhoushan 316021, China
5  Southern Marine Science and Engineering Guangdong Laboratory, Guangzhou 511458, China
*  Correspondence: leiding@mail.sitp.ac.cn; Tel.: +86-571-8196-1201

**Abstract:** The design and processing algorithm of oceanic LIDAR requires an effective lidar simulator. Currently, most simulation methods for lidar signal propagation in seawater use elastic scattering. In this study, a new semi-analytical Monte Carlo (MC) model for oceanic lidar inelastic signals is developed to investigate chlorophyll fluorescence and Raman scattering in seawater. We also used this model to simulate the echo signal of high spectral resolution lidar (HSRL) in the particulate and water molecular channels. Using this model, the effects of chlorophyll concentration, multiple scattering, receiving field of view (FOV), scattering phase function (SPF), receiver full width at half maximum (FWHM) and inhomogeneous seawater were investigated. The feasibility and effectiveness of the model were verified by the lidar equation under small and large FOVs. The results showed that chlorophyll concentration and vertical structure and multiple scattering have considerable and integrated effects on echo signals, which could provide a reference for the design of oceanic fluorescence and HSRL lidar systems and contribute to the development of processing algorithms.

**Keywords:** oceanic lidar; Monte Carlo; inelastic scattering; multiple scattering; inhomogeneous water

## 1. Introduction

Passive ocean-color remote sensing has been used for decades to improve our understanding of the global marine ecosystem and plankton distribution [1]. However, this process has inherent limitations with regard to obtaining information about the vertical structure of seawater and detecting at night and at high latitudes [2]. Oceanic lidar can overcome these limitations by providing the vertical profile of seawater optical properties and subsurface ocean vertical information with high spatial and temporal resolution [3–6]. In addition, as an active remote sensing technology, oceanic lidar can also perform well at night and at high latitudes [7,8].

The design and processing algorithm of oceanic lidar requires an effective atmosphere-ocean radiation transmission model [9,10]. It is thus important to study the parameters of lidar systems, such as multiple scattering, inelastic scattering and SPF, in order to develop lidar radiation transmission models. In recent years, the MC simulation technique has been widely used to study the radiation transmission of oceanic lidar [11,12]. The semi-analytic MC method, which incorporates variance reduction techniques, considers each photon scattering behavior and markedly improves signal convergence speed [9,13]. This method has been verified to be effective via experiments [14]. Therefore, an MC radiation transmission model that is suitable for oceanic lidar should be developed.

Overall, the accurate interpretation of oceanic lidar signals requires a comprehensive understanding of laser propagation processes, including laser crossing the sea–air interface, scattering by underwater phytoplankton and reflection from the seafloor [4]. The seawater backscattering signals received by lidar telescopes contain rich spectral information, such as Mie scattering (particulates), Rayleigh scattering (water molecules), Brillouin scattering (water molecules), Raman scattering (water molecules) and fluorescence [15,16]. Brillouin scattering signals can be used to efficiently determine the temperature, salinity and sound velocity profiles of seawater in real time, giving this method good application potential [17]. In 2010, Asahara et al. used the Brillouin scattering method to determine the speed of sound in water under high temperature and high pressure [18]. In 2018, a team from Nanchang Hangkong University used HSRL to detect objects at a water depth of approximately 65 m [19]. Fluorescence signals are important when studying the content and distribution of plankton and chlorophyll in seawater. In 2020, Matteoli et al. designed a fluorescence lidar simulator but primarily investigated the influence of colored dissolved organic matter (CDOM) without in-depth consideration of factors such as multiple scattering, SPF and instrument parameters [20]. Currently, there is a lack of MC simulation studies for fluorescence signals. Raman scattering signals can be used to extract the backscattering coefficient and analyze the molecular composition of seawater [21]. The Raman scattering intensity of seawater molecules is relatively weak and primarily distributed in a very narrow wavelength range, which requires high resolution and a high signal-to-noise ratio lidar to receive [22]. Some scholars have analyzed the multiple scattering model of Raman scattering echoes [23], but none has applied this method in seawater. Thus, most existing oceanic lidar models can only be used to simulate elastic scattering, and there is a lack of radiation transport simulation studies for inelastic scattering and fluorescence.

In this article, a new model of oceanic lidar radiation transmission is established based on semi-analytical MC simulation. The inelastic scattering including fluorescence, Raman scattering and HSRL signals received by lidar were simulated, and the effects of chlorophyll concentration and FOV on the multiple scattering of each signal were analyzed. Based on the lidar equation, the effectiveness of the model was verified. We also discuss the influence of parameters including SPF, receiver FWHM and inhomogeneous water on the model.

## 2. Materials and Methods

### 2.1. Semi-Analytic MC Model

The MC method can determine the frequency of an event or the average of a random variable experimentally when the probability of a characteristic event occurring or the expectation of a random variable is known for a particular problem [11]. This method is then used as an approximate solution to the problem. Therefore, the MC method is well suited to simulating the radiation transmission of light in water.

We have developed an improved semi-analytical MC simulation model to analytically estimate the probability of photons returning to a remote receiver after inelastic scattering at collision point i. The model is also suitable for inhomogeneous water. For example, a flowchart of the semi-analytical MC lidar echo signal simulation model for fluorescence is shown in Figure 1, where N means No and Y means Yes.

The step size of a photon in the flow chart is a concept that describes the motion of a photon using the MC method. The distance between the two interactions between the photon and the particulate in water is defined as the step size of the photon (i.e., the mean free path of the photon). This quantity is dimensionless and can be expressed as [24]:

$$s = -\frac{1}{c} \ln R \qquad (1)$$

where $c$ is the attenuation coefficient of the beam and $R$ is a random number uniformly distributed between 0 to 1. After each step, the position coordinates and weights of the photons are updated [24].

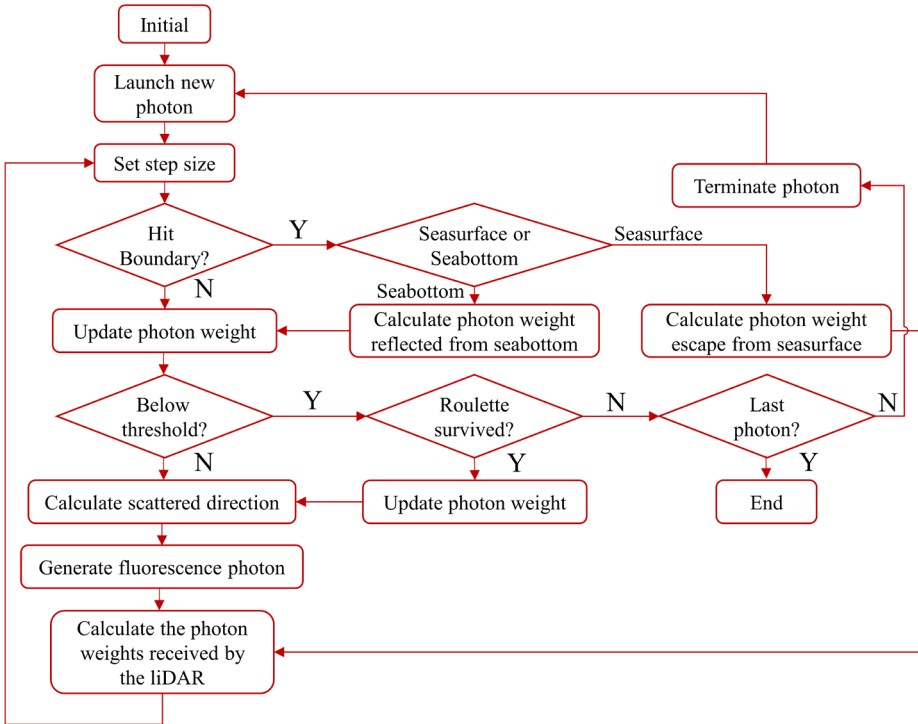

**Figure 1.** Flow chart of the simulation model for the MC lidar fluorescence echo signal.

In this study, an improved semi-analytical MC simulation model was used to improve photon tracking efficiency by treating each photon as a packet of photons. Every time a photon interacts with a particulate or molecule, the probability of the photon packet returning to the detector is directly calculated if the photon is within the action range limited by the field angle of view:

$$E = b(\lambda) \cdot \widetilde{\beta}(\theta) \cdot \Delta\Omega \cdot exp\left[-\sum_{j=1}^{i} c_\lambda(j)d(j)\right] T_a T_t \tag{2}$$

where $b$ is the scattering coefficient; $\widetilde{\beta}(\theta)$ is the SPF; c is the beam attenuation coefficient; $T_a$ is the transmittance of the atmosphere; $T_t$ is the transmittance of the air–sea interface; $d$ is the distance between the current position of the photon and the sea surface along the receiving path; and $\Delta\Omega$ is the solid receiving angle of the receiver aperture seen at the depth of the collision photon, which can be calculated as:

$$\Delta\Omega = \frac{A}{(nH+z)^2} \tag{3}$$

where $A$ is the receiving aperture area of the telescope, $n$ is the refractive index of seawater and $H$ is the height from the lidar to the sea surface. The analytical lidar geometric model of the photon's scattering event is shown in Figure 2.

In this model, when the photon reaches the sea floor, the Lambertian reflectance model is employed for calculating sea bottom reflectance. The reflection of a photon on the Lambertian surface is isotropic, so that the probability of a photon reflected from the seafloor is the same in all directions.

The semi-analytical Monte Carlo method makes full use of every scattering behavior of each photon and markedly improves the convergence speed of the simulation.

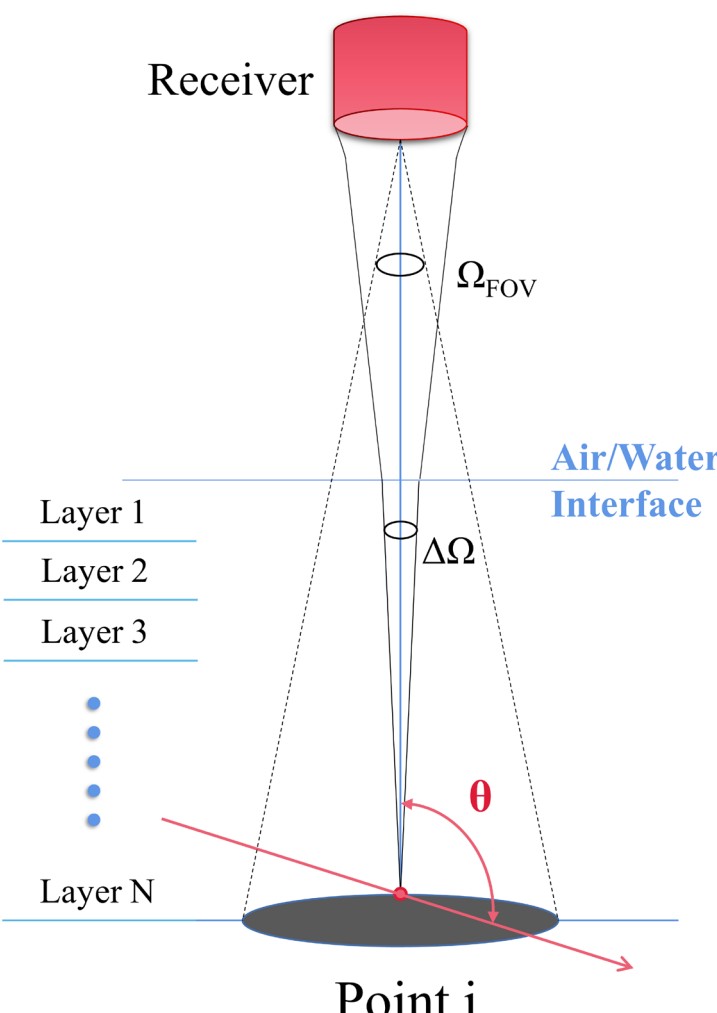

**Figure 2.** Lidar geometric model of photon scatter back to the remote receiver.

### 2.2. HSRL Model

HSRL is a new lidar technology that has been applied in the field of atmospheric remote sensing [25]. However, it was not until 2017 that scholars began to study the feasibility of oceanic HSRL [26]. Oceanic HSRL emits a laser pulse into the sea water and receives the backscattered light from the sea water by a telescope. After collimation, filtering and splitting, the scattered light is detected by the particulate scattering channel and the water molecular scattering channel to analyze the optical properties of the sea water and obtain the scattering and attenuation information of the medium [2].

The HSRL particulate scattering channel primarily receives the Mie scattering signal and Rayleigh scattering, which is a type of elastic scattering. Rayleigh scattering has a relatively weak signal intensity compared to Mie scattering [27]; thus, the particulate scattering channel signal can generally be considered a Mie scattering signal. The Mie scattering coefficient is $b_p$, and the SPF is $\widetilde{\beta}_p(\theta)$. Because particulate scattering does not change the frequency of light, the correlation coefficients are calculated based on the lidar emission wavelength of 532 nm. According to Equation (2), the photon return probability of the HSRL particulate scattering channel is:

$$E_p = b_p(532\ \text{nm}) \cdot \widetilde{\beta}_p(\theta) \cdot \Delta\Omega \cdot exp\left[ -\sum_{j=1}^{i} c_{532nm}(j)d(j) \right] T_a T_t \tag{4}$$

The water molecular scattering channel primarily receives a Brillouin scattering signal, which is a type of inelastic scattering. Brillouin scattering is shifted by 7–8 GHz at 532 nm in the backscattering direction and can be separated by a super narrowband optical spectrum analyzer. The scattering coefficient of Brillouin scattering is $b_w$, and the SPF is $\widetilde{\beta}_w(\theta)$. Because the real wavelength shift of Brillouin scattering is approximately 1.23 nm, the relevant parameters are calculated using 532 nm for ease of calculation. According to Equation (2), the photon return probability of the HSRL particulate scattering channel is:

$$E_w = b_w(532 \text{ nm}) \cdot \widetilde{\beta}_w(\theta) \cdot \Delta\Omega \cdot exp\left[-\sum_{j=1}^{i} c_{532nm}(j)d(j)\right]T_a T_t \tag{5}$$

In general, oceanic HSRL technology is still a relatively novel field, and many related physical processes require additional investigation.

### 2.3. Fluorescence Model

Fluorescence is the light emitted by a substance after it absorbs light or other electromagnetic radiation [28]. When light shines on atoms, the energy of the light makes some electrons around the nuclei jump from their original orbitals to higher-energy orbitals (i.e., from ground state to the first or second excited singlet). The first or second excited singlet is unstable and will return to the ground state [28]. When the electron returns to the ground state from the first excited singlet state, energy is released in the form of light, creating fluorescence [29]. In most cases, the emitted wavelength of light is longer, and the emitted energy is lower than the wavelength of absorption [28].

Fluorescence is the emission of a new photon after absorption, independent of the absorbed photon. However, for most optical oceanographic purposes, particularly to solve time-independent radiation transfer equations, fluorescence can be treated as inelastic scattering and treated in the same mathematical form as the Raman effect. This study focused on chlorophyll fluorescence. The volume inelastic scattering function for chlorophyll fluorescence is [30]:

$$\beta_f(z, \psi, \lambda', \lambda) = b_f(z, \lambda')f_f(\lambda', \lambda)\widetilde{\beta_f}(\theta) \tag{6}$$

where $b_f(z, \lambda')$ is the fluorescence scattering coefficient in units of m$^{-1}$; $f_f(\lambda', \lambda)$ is the fluorescence wavelength redistribution function in units of nm$^{-1}$; and $\widetilde{\beta_f}(\theta)$ is the fluorescence SPF in units of sr$^{-1}$.

For chlorophyll fluorescence, when it is treated as inelastic scattering, the "scattering "coefficient is the absorption coefficient of chlorophyll:

$$b_f(z, \lambda') = Chl(z)a_C^*(\lambda') \tag{7}$$

where $Chl(z)$ is the chlorophyll concentration at depth $z$ and $a_C^*(\lambda')$ is the chlorophyll-specific absorption spectrum [30].

The fluorescence emission spectrum is independent of the excitation wavelength, which means that any photon with a wavelength in the range of 370 to 690 nm that is absorbed leads to the same fluorescence [31]. The fluorescence wavelength redistribution function can be written as:

$$f_f(\lambda', \lambda) = \Phi_f g_f(\lambda')h_f(\lambda)\frac{\lambda'}{\lambda} \tag{8}$$

where $\Phi_f$ is the quantum efficiency for chlorophyll fluorescence; $g_f(\lambda')$ specifies the interval over which light can excite fluorescence; and $h_f(\lambda)$ is the fluorescence wavelength emission function. Falkowski et al. identified 200,000 profiles of surface waters in different locations and reported an average of $\Phi_f = 0.07$ [32].

As mentioned earlier, incident light in the wavelength range of 370 to 690 nm, if absorbed, may also excite fluorescence. Therefore, $g_f(\lambda')$ can be described as:

$$g_f(\lambda') = \begin{cases} 1 & if \ 370 \leq \lambda' \leq 690 \ \mathrm{nm} \\ 0 & otherwise \end{cases} \tag{9}$$

The emission function $h_f(\lambda)$ is commonly approximated as a Gaussian [33]:

$$h_f(\lambda) = \frac{1}{\sqrt{2\pi}\sigma_f} exp\left[-\frac{1}{2}\left(\frac{\lambda - \lambda_f}{\sigma_f}\right)^2\right] \tag{10}$$

where $\lambda_f = 685$ nm is the wavelength of maximum emission and $\sigma_f$ is the standard deviation of the Gaussian, which corresponds to the FWHM of the fluorescence excitation spectrum $FWHM_{FLUOR}$:

$$\sigma_f = \frac{FWHM_{FLUOR}}{2\sqrt{2\ln 2}} \tag{11}$$

As mentioned before, the emission of fluorescence is isotropic. Therefore, the phase function can be simply described as [30]:

$$\widetilde{\beta_f}(\theta) = \frac{1}{4\pi} \tag{12}$$

According to Equation (2), the photon return probability of the fluorescence signal is:

$$E_f = \frac{b_f(z, 532 \ \mathrm{nm}) f_f(532 \ \mathrm{nm}, 685 \ \mathrm{nm})}{4\pi} \cdot \Delta\Omega \cdot exp\left[-\sum_{j=1}^{i} c_{685 \ \mathrm{nm}}(j)d(j)\right] T_a T_t \tag{13}$$

### 2.4. Raman Scattering Model

Raman scattering is a type of inelastic scattering of light by molecules [34]. Incident light can excite a molecule in its ground state to a higher energy level, which then immediately decays back to a lower level after the emission of light [30,34]. If the decay returns the molecule to its initial state, the scattering is elastic and is called Rayleigh scattering. If the decay is to a molecular vibrational level above the ground state, then emitted light has a longer wavelength (lower energy) than the incident light; this process describes Raman scattering [35].

Raman scattering in water creates a large wavelength shift, corresponding to a wavenumber shift of approximately 3400 cm$^{-1}$, which is many tens to more than 100 nanometers at visible wavelengths [30]. The time scale of Raman scattering is approximately $10^{-13}$ to $10^{-12}$ s [34], which is much faster than the time scale of fluorescence. The incorporation of Raman scatter into radiative transfer calculations is:

$$\beta_R(\lambda', \lambda) = b_R(\lambda') f_R(\lambda', \lambda) \widetilde{\beta_R}(\theta) \tag{14}$$

where $b_R(\lambda')$ is the Raman scattering coefficient in units of m$^{-1}$; $f_R(\lambda', \lambda)$ is the Raman wavelength distribution function in units of nm$^{-1}$; $\widetilde{\beta_R}(\theta)$ is the Raman SPF in units of sr$^{-1}$.

The Raman scattering coefficient $b_R(\lambda')$ represents how much of the excited irradiance is scattered to all emission wavelengths $\lambda > \lambda'$ per unit distance at the excited wavelength $\lambda'$. The most recently published values of $b_R(488 \ \mathrm{nm})$ for water are $2.4 \times 10^{-4}$m$^{-1}$ [36]. In terms of excitation wavelength, Bartlett et al. found a formula that can be used for energy calculation [37]:

$$b_R(\lambda') = b_R(488 \ \mathrm{nm})(488/\lambda')^{5.5 \pm 0.4} \tag{15}$$

The Raman wavelength distribution function $f_R(\lambda', \lambda)$ is related to the excitation wavelength and emission wavelength. This function is most conveniently described by

the corresponding wavenumber distribution function $f_R(k'')$, where $k''$ is the wavenumber shift and expressed in units of cm$^{-1}$. The $f_R(k'')$ for water is given by a sum of four Gaussian functions [38]:

$$f_R(k'') = \left[ \left( \frac{\pi}{4\ln 2} \right)^{\frac{1}{2}} \sum_{i=1}^{4} R_i \right]^{-1} \sum_{i=1}^{4} R_j \frac{1}{\triangle k_j} exp \left[ -4\ln 2 \frac{(k'' - k_j)^2}{\triangle k_j^2} \right] \tag{16}$$

where $k_j$ is the center of the $j^{th}$ Gaussian function in units of cm$^{-1}$; $\triangle k_j$ is the full width at half maximum of the $j^{th}$ Gaussian function in units of cm$^{-1}$; and $R_j$ is the nondimensional weight of the $j^{th}$ Gaussian function. $R_j$, $k_j$ and $\triangle k_j$ for pure water at a temperature of 25 °C are shown in Table 1. For water, the wavenumber shift $k''$ is approximately 3400 cm$^{-1}$.

**Table 1.** Parameters for the Raman wavenumber redistribution function $f_R(k'')$ [38].

| $j$ | $R_j$ | $k_j$ | $\triangle k_j$ |
|---|---|---|---|
| 1 | 0.41 | 3250 | 210 |
| 2 | 0.39 | 3425 | 175 |
| 3 | 0.10 | 3530 | 140 |
| 4 | 0.10 | 3625 | 140 |

The Raman phase function $\widetilde{\beta_R}(\theta)$ yields the angular distribution of the Raman scattering. This function can be described as:

$$\widetilde{\beta_R}(\theta) = \frac{3}{16\pi} \frac{1+3\rho}{1+2\rho} \left[ 1 + \left( \frac{1-\rho}{1+3\rho} \right) cos^2\theta \right] \tag{17}$$

where $\theta$ is the scattering angle between the direction of the incident and scattered radiance and $\rho$ is the depolarization factor. The latter depends on the wavenumber shift [38]. For a value of $k'' = 3400$ cm$^{-1}$, $\rho \approx 0.18$.

According to Equation (2), the photon return probability of the Raman scattering signal is:

$$E_R = b_R(532 \text{ nm}) f_R(532 \text{ nm}, 650 \text{ nm}) \widetilde{\beta_R}(\theta) \cdot \Delta\Omega \cdot exp \left[ -\sum_{j=1}^{i} c_{650 \text{ nm}}(j) d(j) \right] T_a T_t \tag{18}$$

### 2.5. Hydrosol Model

The basic optical properties of seawater are divided into Inherent Optical Properties (IOP) and Apparent Optical Properties (AOP). The IOP only depends on the composition of the water, including refractive index n, absorption coefficient $a$ (m$^{-1}$), scattering coefficient $b$ (m$^{-1}$), beam attenuation coefficient $c$ (m$^{-1}$), scattering phase function $\widetilde{\beta}$ (sr$^{-1}$) and backscattering coefficient $b_b$ (m$^{-1}$). Among them, the beam attenuation coefficient is equal to the sum of the absorption coefficient and the scattering coefficient ($c = a + b$).

The IOP of seawater is determined by the composition of the substances it contains. Seawater is a complex multicomponent aqueous solution, among which the primary components that affect the optical properties of seawater are pure seawater, phytoplankton, CDOM, non-phytoplankton organic particulates (NAP), inorganic particulates and bubbles. This study focuses on case-1 water, whose optical properties are primarily determined by pure water and phytoplankton [39].

The absorption coefficient of seawater is defined as the spectral absorption rate per unit distance in the seawater medium, which is strongly spectrally dependent. The magnitude of the absorption coefficient varies with the concentration of the absorbing material.

Theoretically, the absorption coefficient can be expressed as the sum of the absorption coefficients of each component [40]:

$$a(\lambda) = a_w(\lambda) + a_p(\lambda) \tag{19}$$

where subscripts $w$ and $p$ indicate water and phytoplankton, respectively. In the model, we interpolated the measured data to obtain the absorption coefficients under different wavelength and chlorophyll concentration [39]. The absorption coefficient of water is lower in blue and increases into the ultraviolet, red and infrared. Chlorophyll typically has two absorption peaks at the blue and red spectrum due to its structure [30].

For coastal waters, the total absorption coefficient includes seawater, particles and colored dissolved organic matter (CDOM), as follows:

$$a(\lambda) = a_w(\lambda) + a_p(\lambda) + a_g(\lambda) \tag{20}$$

where $a_w(\lambda)$ is the absorption coefficient of pure water that has been measured in [41]. $a_p(\lambda)$ is the absorption coefficient of the particles and is mainly contributed by chlorophyll-a. It can be estimated by [42]:

$$a_p(\lambda) = A(\lambda)(Chl)^{E(\lambda)} \tag{21}$$

where $A(\lambda)$ is the specific absorption coefficient and $E(\lambda)$ is the index absorption coefficient. The used data come from the Ocean Optics Web Book [43]. $a_g(\lambda)$ is the absorption coefficient of CDOM, which can be estimated as follows [44]:

$$a_g(\lambda) = a_g(\lambda_0) \times exp(-S(\lambda - \lambda_0)) \tag{22}$$

where $S$ is the spectral absorption slope and $a_g(\lambda_0)$ is the absorption coefficient of CDOM at the reference wavelength $\lambda_0$.

The scattering coefficient b of seawater represents the spectral scattering rate per unit distance in seawater. The biological-optical model of the case 1 water scattering coefficient is [39]:

$$b(\lambda) = b_w(\lambda) + b_p(\lambda) \tag{23}$$

$$b_w(\lambda) = 16.06 \times \left(\frac{550}{\lambda}\right)^{4.324} \times 1.21 \times 10^{-4} , \ b_p(\lambda) = \frac{550}{\lambda} \times 0.3 \times Chl^{0.62} \tag{24}$$

where *Chl* is the concentration of chlorophyll.

Considering the directional characteristics of scattering, the spectral scattering coefficient $b(\lambda)$ can be expressed as the integral form of the spectral volume scattering function $\beta(\theta, \lambda)$ in all directions:

$$b(\lambda) = 2\pi \int_0^\pi \beta(\theta, \lambda) \sin\theta d\theta \tag{25}$$

where the spectral volume scattering function $\beta(\theta, \lambda)$ describes the scattering intensity per unit solid angle per unit distance.

The spectral SPF $\widetilde{\beta}(\theta, \lambda)$ of seawater is defined as:

$$\widetilde{\beta}(\theta, \lambda) = \frac{\beta(\theta, \lambda)}{b(\lambda)} \tag{26}$$

The total scattered phase function $\widetilde{\beta}(\theta)$ is composed of the pure water phase function $\widetilde{\beta}_w(\theta)$ and the particulate phase function $\widetilde{\beta}_p(\theta)$, where $\widetilde{\beta}_w(\theta)$ is:

$$\widetilde{\beta}_w(\theta) = 0.06225\left(1 + 0.835cos^2\theta\right) \tag{27}$$

Due to the limitation of instruments, the particulate SPF $\widetilde{\beta}_p(\theta)$ is difficult to obtain [45]. A simple analytic formula is generally used to approximate the real phase function [46], the most common being the Henyey-Greenstein (HG) function [47]:

$$\widetilde{\beta}_p(\theta) = \widetilde{\beta}_{HG}(\theta, g) = \frac{1}{4\pi} \frac{1 - g^2}{(1 + g^2 - 2g cos\theta)^{3/2}} \tag{28}$$

where $g$ is the asymmetric coefficient that determines the relative magnitude of forward and backscattering in the phase function. This term is defined as:

$$g = 2\pi \int_0^\pi \widetilde{\beta}(\theta, \lambda) cos\theta \sin\theta d\theta = 0 \tag{29}$$

When $g = 0.924$, the particulate phase function measured by Petzold can be well fitted [48]. In this case, the HG phase function can better describe the forward scattering characteristics of particulates in seawater. Other functions, such as TTHG functions, ModHG functions and FF functions, can also be used to simulate phase functions [48,49].

The AOP depends on the inherent optical properties of seawater and on the distribution of the underwater radiation field, including spectral reflectance $R_{rs}$ and diffuse attenuation coefficient $K_d(m^{-1})$.

$K_d$ is an AOP that is commonly used in passive water color remote sensing and is an important AOP used to describe the attenuation coefficient of lidar in oceanic lidar remote sensing. The bio-optical model of $K_d$ in case-1 water is [50]:

$$K_d = a_w + \frac{b_w}{2} + 0.04826 Chl^{0.67224} \tag{30}$$

These parameters have been widely used in simulating ocean exploration and are important to describe the optical properties of seawater and to simulate the radiation transmission of lidar in seawater.

### 2.6. LUT Method for Arbitrary SPF

In MC simulation, SPF is a key parameter to simulate angle scattering. HG SPF was widely used in most previous studies [47] but was not ideal at large or small scattering angles and did not match measured values accurately [46,51,52]. Chen et al. developed a lookup table (LUT) method to establish a LUT of $\theta$ versus $F(\theta)$ [24], which can determine the scattering angle of SPF $\widetilde{\beta}(\theta)$, where $F(\theta)$ is the cumulative distribution function (CDF) of $\widetilde{\beta}(\theta)$. The LUT method has been shown to be suitable for various complex SPFs (e.g., Fournier-Forand phase function (FF) [53], two-term HG phase function (TTHG) [54], Petzold's phase function [55]).

Using MC to simulate light propagation in water requires random selection of photon step size and scattering angle. Generally, these random variables are assigned a random number between 0 to 1. The scattered phase function is a probability density function that describes the angular distribution of light scattered by suspended particulates toward the direction of $\theta$ [4]. $F(\theta)$ is the CDF of the SPF within the range of between 0 to 1; thus, $\theta$ can be solved by random numbers. The CDF of SPF can be obtained from the following equation:

$$F(\theta_i) = 2\pi \sum_{n=1}^{n=i} \widetilde{\beta}(\theta_n) \sin(\theta_n) \Delta\theta_n \tag{31}$$

where $n$ is the index for discrete angle tabulated data, $\theta_n$ is the angle at the index $n$ and $\Delta\theta_n$ is the angle interval between the angles from $n-1$ to $n$. For the requirement of integral operation, $F(\theta_n)$ is the discrete form for $F(\theta)$.

In MC randomization, $F(\theta)$ is equal to a random number $\xi$ ranging between 0 and 1. Thus, we can use $\xi$ to determine the scattering angle $\theta$. When given a random variable $\xi$, if its value meets $F(\theta_{n-1}) < \xi < F(\theta_n)$, then the scattering angle can be obtained.

### 3. Results

#### 3.1. Effect of Different Chlorophyll Concentrations

The effect of different chlorophyll concentrations in homogeneous water on the simulation was analyzed. The concentration of chlorophyll affects the absorption coefficient a and scattering coefficient b of seawater, as shown in Figure 3a. The excitation of fluorescence is also related to chlorophyll.

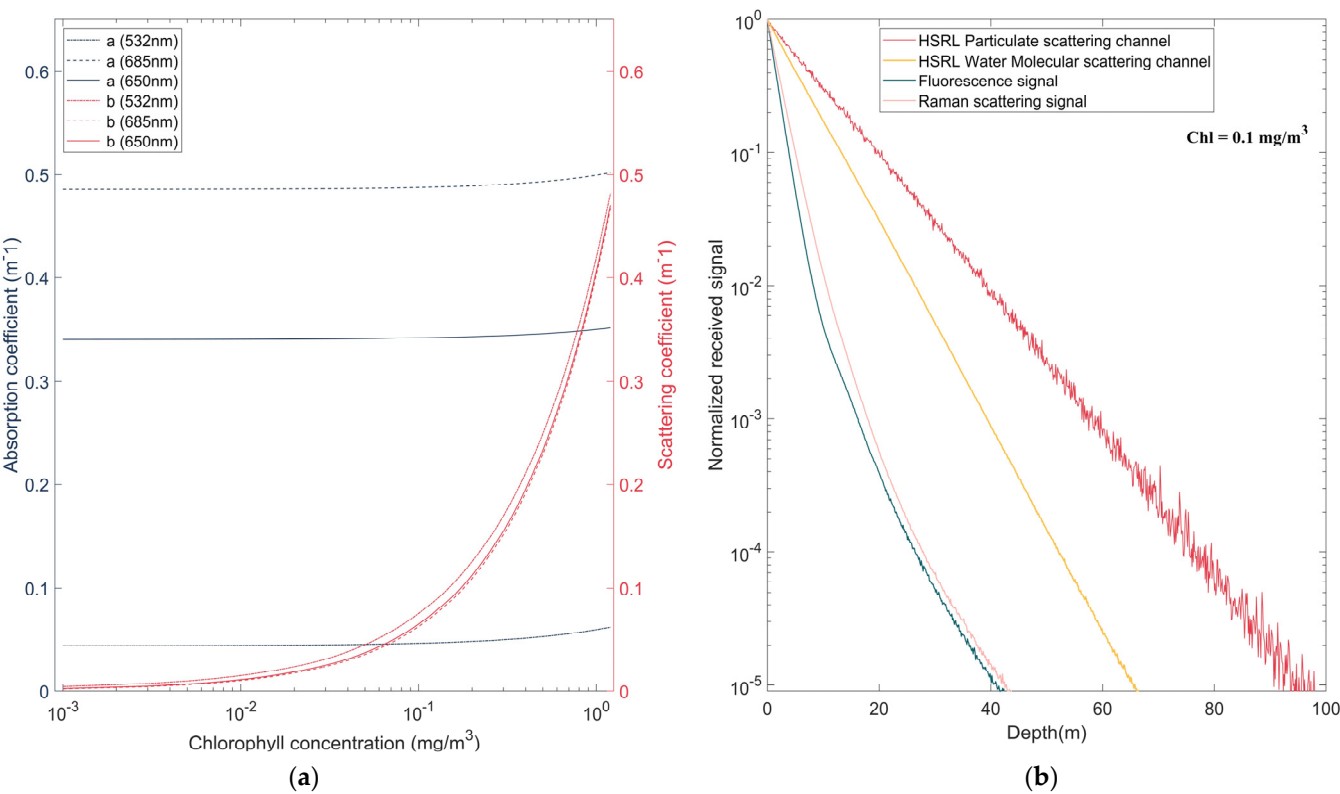

**Figure 3.** (**a**) Variation in absorption coefficient a and scattering coefficient b with chlorophyll concentration. (**b**) Normalized signals of HSRL, fluorescence and Raman scattering at a chlorophyll concentration of 0.1 mg/m$^3$.

Simulations were conducted using 100 million photons for a given detector height of $H$ = 150 m, a field of view (FOV) = 0.2 rad, a detector aperture of $A$ = 0.06 m$^2$, a sea bottom depth of 100 m and a bottom albedo of $\rho_b$ = 0.02. The dynamic range of the receiver was assumed to be 100 dB. The simulation used the HG function, and the scattering asymmetry Factor g was set as 0.924, which was the mean of the scattering angle cosine of the SPF, highlighting the asymmetry between forward scattering and back scattering. The echo signals of the particulate channel and water molecule channel of HSRL lidar, chlorophyll fluorescence and Raman scattering were also simulated.

Figure 3b shows the normalized curves of each echo signal when the chlorophyll concentration is 0.1 mg/m$^3$. The fluorescence and Raman scattering signals decay relatively quickly with depth, and the signal is beyond the dynamic range of the receiver when the depth is approximately 40 m. The HSRL normalized signal changes roughly linearly with depth. Due to the higher scattering coefficient, the particulate channel attenuation is relatively slow and can receive signals up to 100 m deep. Because the HG function is used as the SPF, the particulate channel signal exhibits large fluctuations, while the other signals are relatively smooth.

Four chlorophyll concentrations were then selected for simulation: 0.002, 0.01, 0.1 and 1 mg/m$^3$. The simulation running time under different chlorophyll concentrations and different scattering types is shown in Figure 4. The difference in running time occurs

because with a higher the concentration of chlorophyll in seawater, more multiple scattering occurs, which increases the calculation of multiple propagation paths [4].

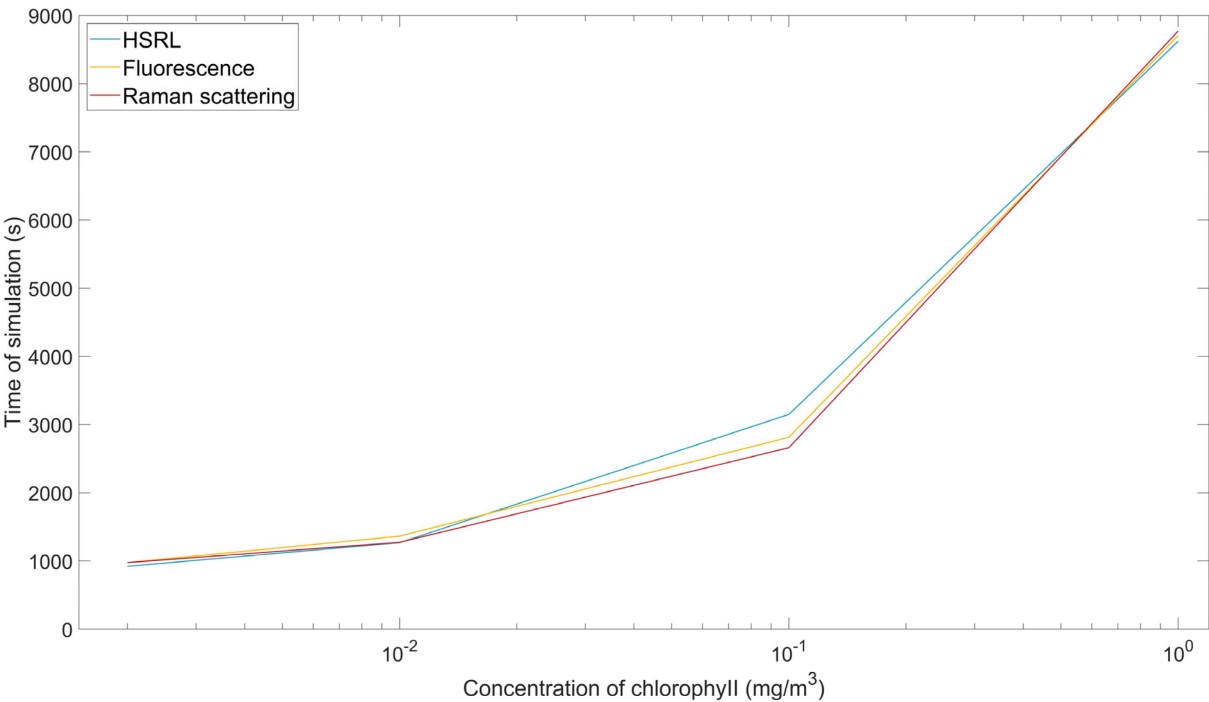

**Figure 4.** Time required to simulate HSRL, fluorescence and Raman scattering under different chlorophyll concentrations.

As shown in Figure 5a,b, with increasing chlorophyll concentration in seawater, the HSRL particulate scattering channel signal and water molecular scattering channel signal intensity decrease rapidly. This result indicates that with increasing water depth, more noise is generated in muddy water due to multiple scattering than in clear water [56]. As shown in Figure 5c, the intensity of the fluorescence echo signal in the shallow water area (depth less than 8.5 m) decreases rapidly with increasing seawater chlorophyll concentration, similar to the HSRL signal. When the depth is greater than 8.5 m, the intensity of the echo signal decreases more slowly with increasing chlorophyll concentration because the fluorescence excitation intensity is proportional to the chlorophyll concentration and multiple scattering increases.

Raman scattering is the inelastic scattering of incident light by water molecules. Although it is not directly related to chlorophyll concentration, the attenuation coefficient of the signal in seawater is affected by the latter. This results in the difference in the echo signal under different chlorophyll concentrations. The echo signal of Raman scattering is similar to that of fluorescence. When the depth is greater than 13.5 m, the echo signal drops more slowly at high chlorophyll concentrations due to multiple scattering, as shown in Figure 5d.

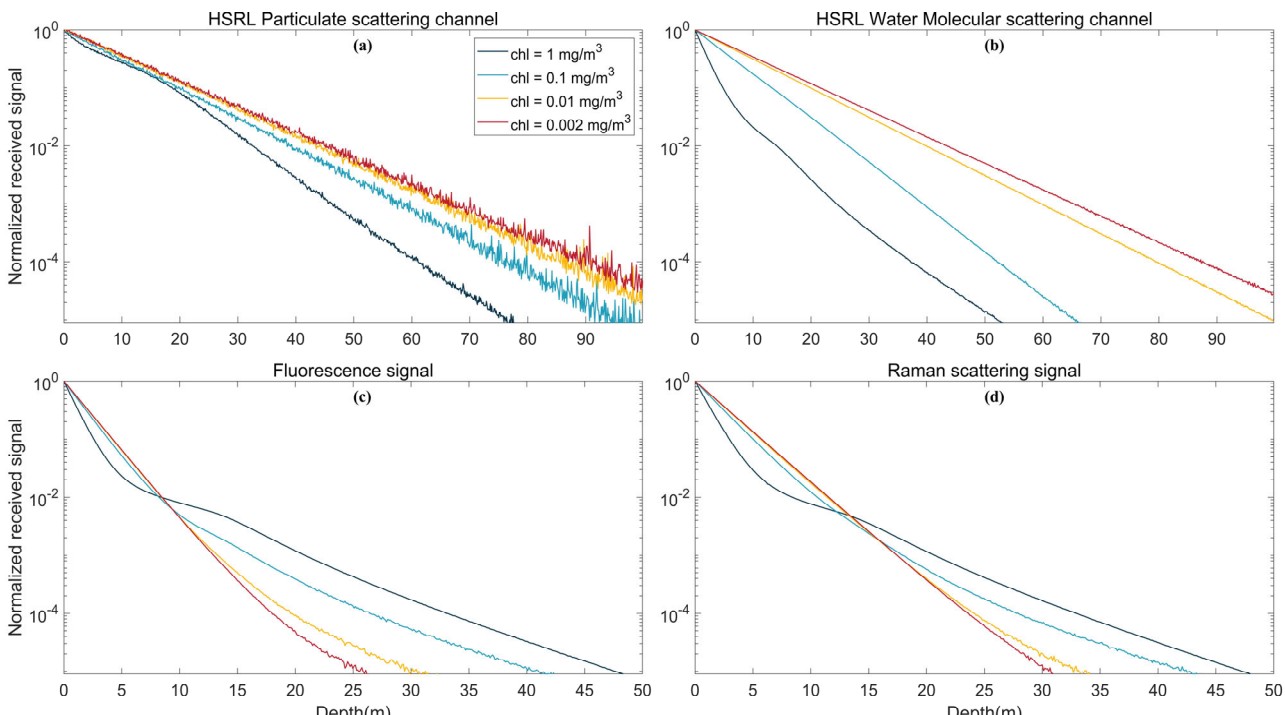

**Figure 5.** Echo signal under different chlorophyll concentrations: (**a**) HSRL particulate scattering channel; (**b**) HSRL water molecular scattering channel; (**c**) fluorescence signal; and (**d**) Raman scattering signal.

### 3.2. Effect of Multiple Scattering

Multiple scattering is an important factor that must be considered in oceanic lidar signal simulation. We simulated single, double, three and more than three scatterers to better understand the effects of multiple scatterers. The lidar system parameter settings are the same as in Section 3.1. As shown in Figure 6, the multiple scattering increases with depth from 0 to 5 m because when the laser enters the water, the wide part of the backscattering in the telescope field of view is single scattering, and the multiple scattering signal is nearly zero. With increasing depth, the frequency of multiple scattering increases, and the intensity of the multiple scattering signal increases. However, although the amount of multiple scattering increases, the signal amplitude decreases with increasing depth due to the attenuation of water and the fact that most of the backward signal is outside the telescope's FOV.

Many researchers [12,57] have proposed that the effective attenuation coefficient applicable to lidar systems must fall between the beam attenuation coefficient c and diffuse attenuation coefficient $K_d$. As shown in Figure 7a,b, the slope of the returned HSRL signal (Lidar effective attenuation coefficient, $K_{sys}$) approaches c in a narrow FOV because nearly all of the scattered photons are not picked up by the telescope [58]. When the field of view is sufficiently large, $K_{sys}$ approaches $K_d$ because the lost portion of the photons received by the telescope are those absorbed by seawater. These results are consistent with previous studies [56,59].

For the fluorescence and Raman scattering signal, the signal in the shallow water area (depth less than 10 m) is also in good agreement with the above law. However, for a large FOV system with increasing depth, the difference between the echo signal amplitude and the signal amplitude calculated by the lidar equation becomes increasingly larger (Figure 7c,d). This result shows that the method of retrieving fluorescence and Raman scattering signals by lidar equations alone is not suitable. The effects of multiple scattering under different FOVs are discussed in detail in Section 4.

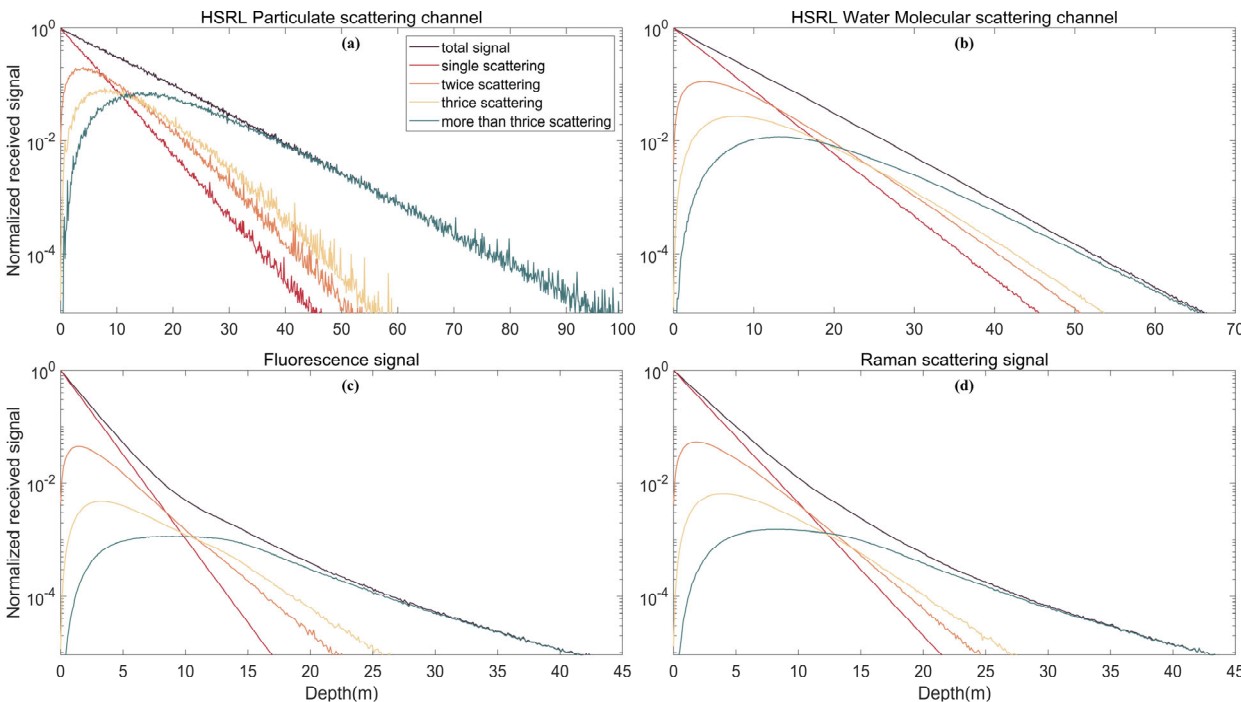

**Figure 6.** Multiple scattering signals of different echo signals: (**a**) HSRL particulate scattering channel; (**b**) HSRL water molecular scattering channel; (**c**) fluorescence signal; and (**d**) Raman scattering signal.

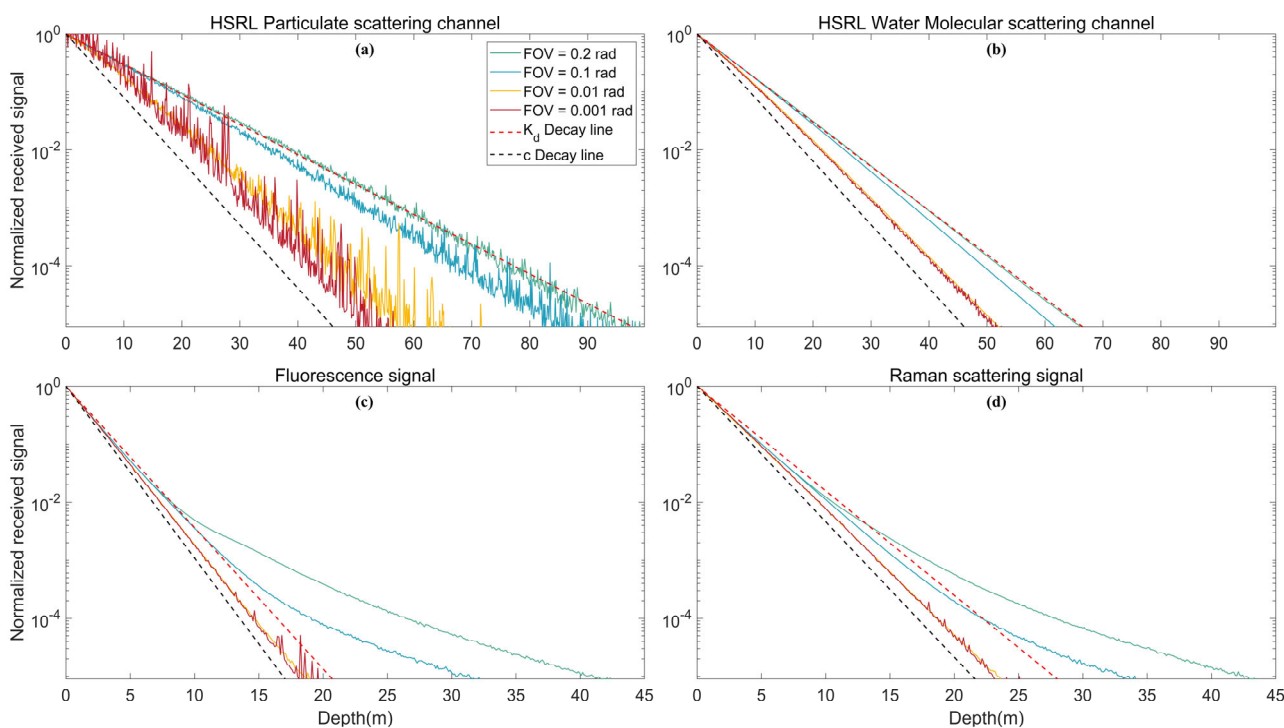

**Figure 7.** Comparison of lidar equations with multiple scattered signals under different FOVs: (**a**) HSRL particulate scattering channel; (**b**) HSRL water molecular scattering channel; (**c**) fluorescence signal; and (**d**) Raman scattering signal.

## 4. Discussion

### 4.1. Multiple Scattering Contributions under Different FOVs

In Section 3, we discovered that multiple scattering is an important factor in lidar system design. Simulation of multiple scattering contributions under different FOVs is required to guide the setting of lidar instrument parameters. Therefore, we analyze the effects of multiple scattering on inelastic scattered signals under different FOVs. The parameter settings of the lidar system are the same as those in Section 3.

Figures 8 and 9 show the simulated signals of the HSRL particulate scattering channel, water molecular scattering channel, fluorescence and Raman scattering under different FOVs of 0.2, 0.1, 0.01 and 0.001 rad. Results show that the multiple scattering contribution increases with increasing depth. At a small FOV (0.001 rad), multiple scattering is negligible, and only single scattering is generally considered. At this point, because the FOV is small, most echo signals are not received, resulting in distortion of the signal curve and information loss. For a large FOV, after the depth reaches a "certain depth", the echo energy is primarily contributed by multiple scattering, which cannot be calculated by the lidar equation. The "certain depth" decreases as the FOV increases. Compared with fluorescence and Raman scattering signals, this multiple scattering weight change of HSRL molecular channel signals (mainly Brillouin scattering) is small under different FOVs. For a larger FOV, lidar can detect deeper water due to the contribution of multiple scatters. However, increasing the FOV also introduces more background stray light; thus, there are trade-offs. In general, oceanic remote sensing, 0.1 rad is the appropriate FOV. Thus, we cannot ignore multiple scattering effects [60].

### 4.2. Multiple Scattering Contributions under Different Chlorophyll Concentrations

The effect of multiple scattering on the inelastic scattering signal at different chlorophyll concentrations was analyzed. The lidar system parameter settings are the same as in each subsection of Section 3. The results in Figures 10 and 11 show that with an increase in chlorophyll concentration, the weight of multiple scattering in the total signal also increases. This relationship occurs because a higher concentration of chlorophyll leads to more particulates in the seawater, and multiple scattering is more likely to occur. At low chlorophyll concentrations, the single scattering signal is dominant. However, at a high chlorophyll concentration (1 mg/m$^3$), multiple scattering contributed most of the echo signal after the depth increased to more than 10 m. This process also results in a nonlinear decrease in fluorescence and Raman scattering signals with increasing depth. Fluorescence and Raman multiple scattering signals have stronger effects on the total signal than the HSRL signal.

For fluorescence and Raman scattering, the higher the chlorophyll concentration, the deeper the echo signal that can be received in the telescopic dynamic range, which is also one of the contributions of multiple scattering. On the other hand, for HSRL particulate scattering channel and water molecular scattering channel, the conclusion is the opposite. This is because the effect of increased attenuation coefficient caused by higher chlorophyll concentration is greater than the effect of multiple scattering.

Therefore, combined with the analysis in Section 4.1, clear water requires a large FOV to receive multiple scattered signals, while turbid water requires a small FOV to receive a single scattered signal. The multiple scattering signals of the four signals were relatively similar at each chlorophyll concentration.

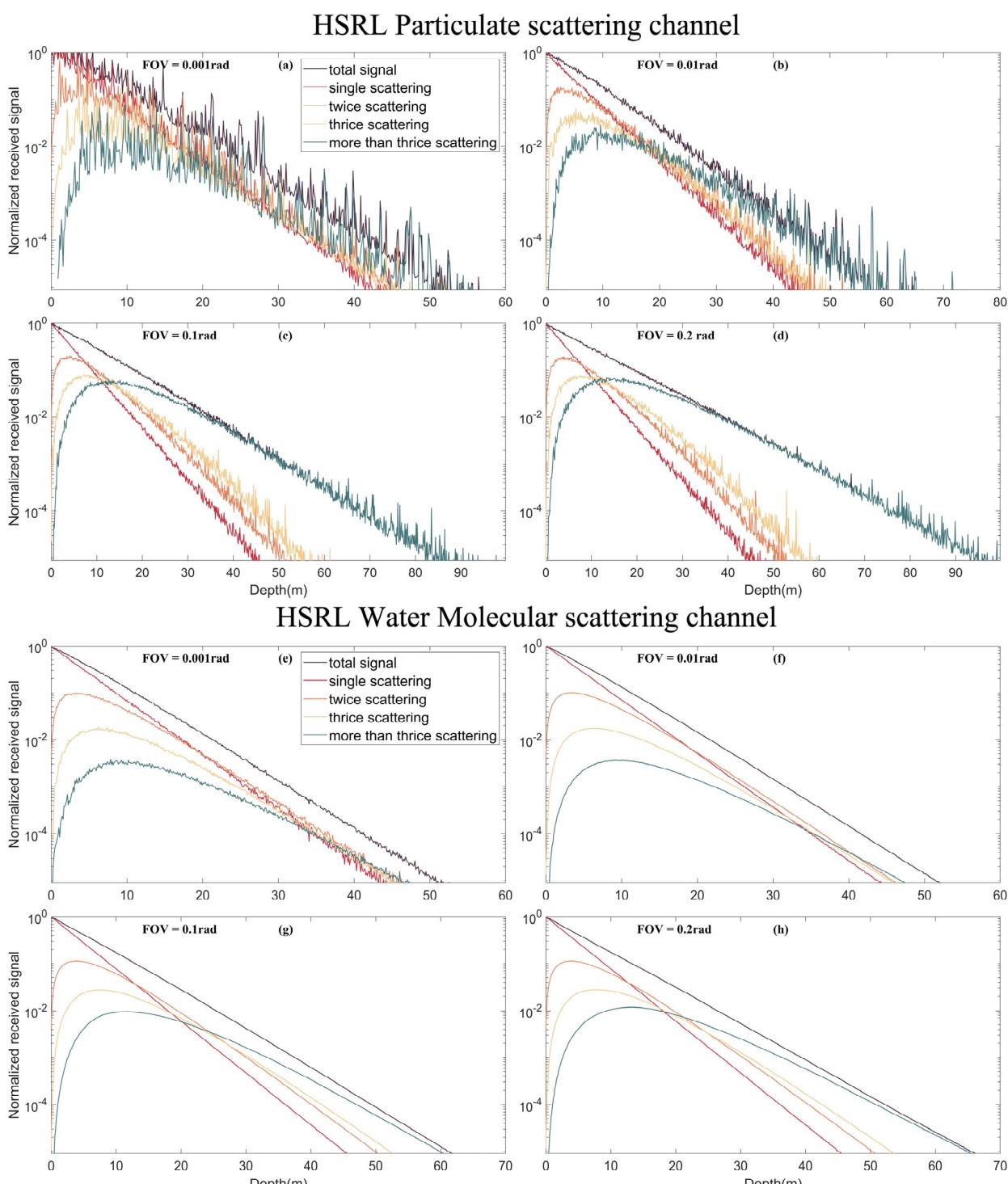

**Figure 8.** Multiple scattering signals of different echo signals under different FOVs. (**a**) Multiple scattering signals in the HSRL particle scattering channel under FOV of 0.001 rad; (**b**) Multiple scattering signals in the HSRL particle scattering channel under FOV of 0.01 rad; (**c**) Multiple scattering signals in the HSRL particle scattering channel under FOV of 0.1 rad; (**d**) Multiple scattering signals in the HSRL particle scattering channel under FOV of 0.2 rad; (**e**) Multiple scattering signals in the HSRL water molecular scattering channel under FOV of 0.001 rad; (**f**) Multiple scattering signals in the HSRL water molecular scattering channel under FOV of 0.01 rad; (**g**) Multiple scattering signals in the HSRL water molecular scattering channel under FOV of 0.1 rad; (**h**) Multiple scattering signals in the HSRL water molecular scattering channel under FOV of 0.2 rad.

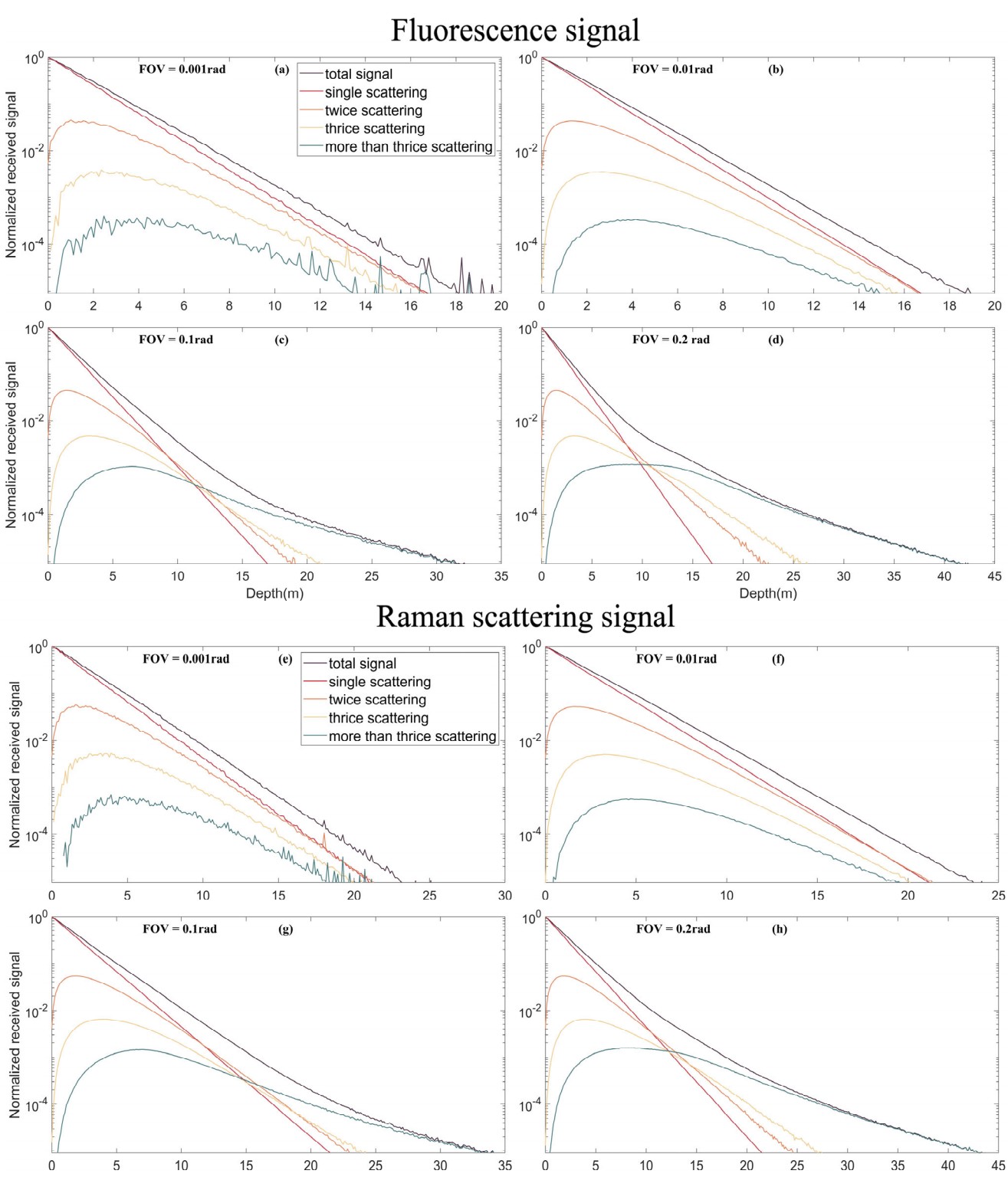

**Figure 9.** Multiple scattering signals of different echo signals under different FOVs. (**a**) Multiple scattering signals in the fluorescence under FOV of 0.001 rad; (**b**) Multiple scattering signals in the fluorescence under FOV of 0.01 rad; (**c**) Multiple scattering signals in the fluorescence under FOV of 0.1 rad; (**d**) Multiple scattering signals in the fluorescence under FOV of 0.2 rad; (**e**) Multiple scattering signals in the Raman scattering under FOV of 0.001 rad; (**f**) Multiple scattering signals in the Raman scattering under FOV of 0.01 rad; (**g**) Multiple scattering signals in the Raman scattering under FOV of 0.1 rad; (**h**) Multiple scattering signals in the Raman scattering under FOV of 0.2 rad.

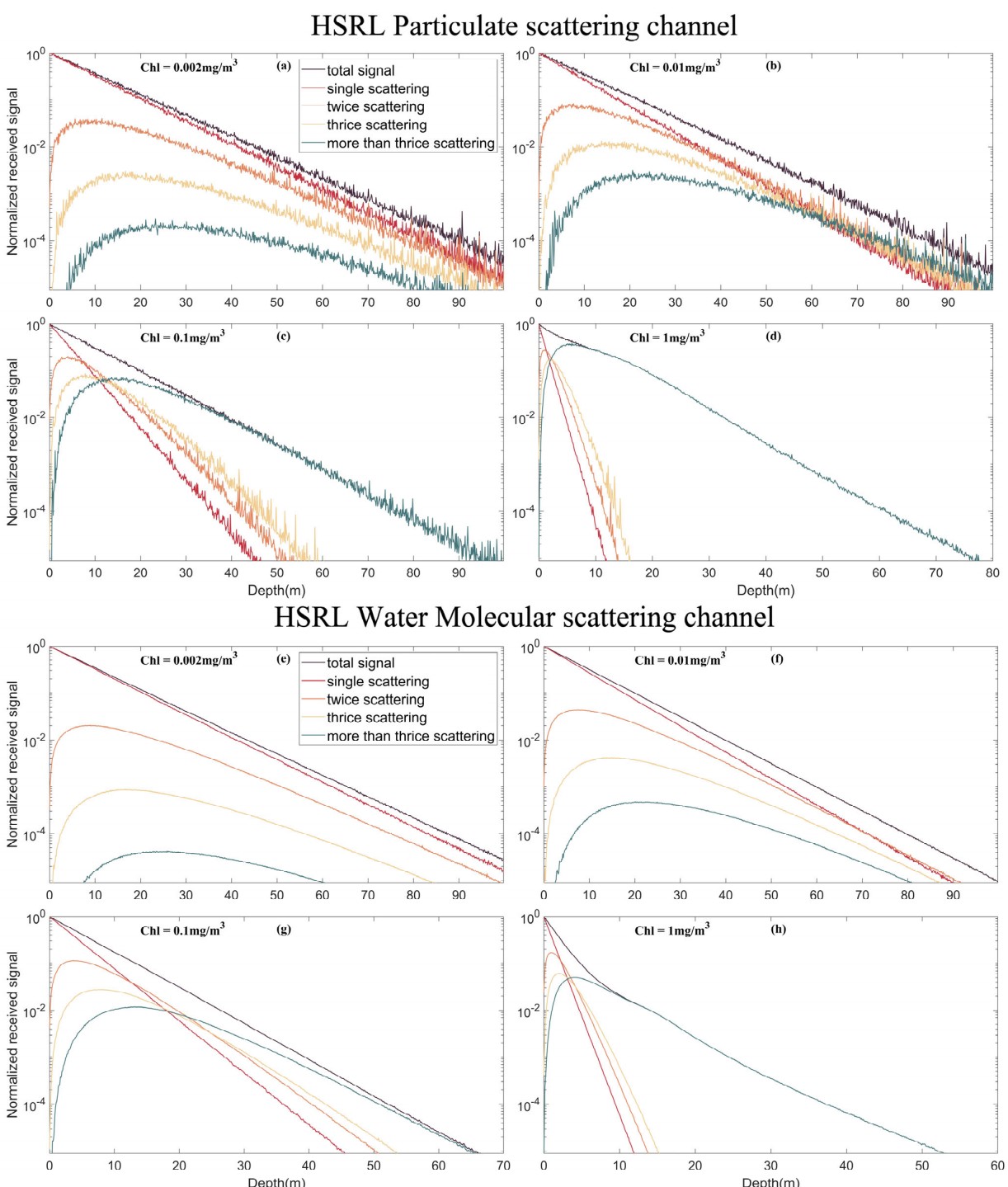

**Figure 10.** Multiple scattering signals of different echo signals under different chlorophyll concentrations. (**a**) Multiple scattering signals in the HSRL particle scattering channel under chlorophyll concentrations of 0.002 mg/m$^3$; (**b**) Multiple scattering signals in the HSRL particle scattering channel under chlorophyll concentrations of 0.01 mg/m$^3$; (**c**) Multiple scattering signals in the HSRL particle scattering channel under chlorophyll concentrations of 0.1 mg/m$^3$; (**d**) Multiple scattering signals in the HSRL particle scattering channel under chlorophyll concentrations of 1 mg/m$^3$; (**e**) Multiple scattering signals in the HSRL water molecular scattering channel under chlorophyll concentrations of 0.002 mg/m$^3$; (**f**) Multiple scattering signals in the HSRL water molecular scattering channel under chlorophyll concentrations of 0.01 mg/m$^3$; (**g**) Multiple scattering signals in the HSRL water molecular scattering channel under chlorophyll concentrations of 0.1 mg/m$^3$; (**h**) Multiple scattering signals in the HSRL water molecular scattering channel under chlorophyll concentrations of 1 mg/m$^3$.

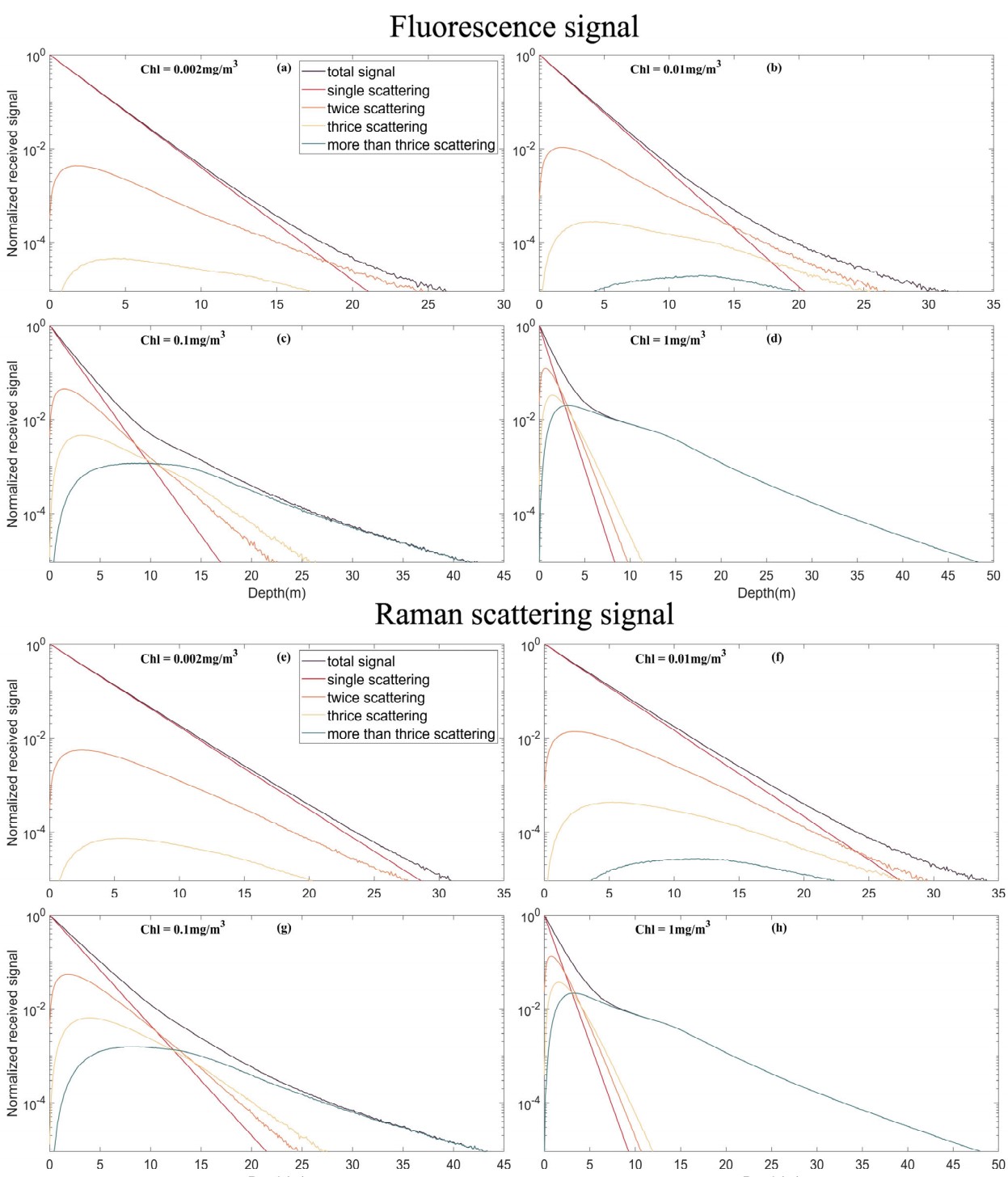

**Figure 11.** Multiple scattering signals of different echo signals under different chlorophyll concentrations. (**a**) Multiple scattering signals in the fluorescence under chlorophyll concentrations of 0.002 mg/m³; (**b**) Multiple scattering signals in the fluorescence under chlorophyll concentrations of 0.01 mg/m³; (**c**) Multiple scattering signals in the fluorescence under chlorophyll concentrations of 0.1 mg/m³; (**d**) Multiple scattering signals in the fluorescence under chlorophyll concentrations of 1 mg/m³; (**e**) Multiple scattering signals in the Raman scattering under chlorophyll concentrations of 0.002 mg/m³; (**f**) Multiple scattering signals in the Raman scattering under chlorophyll concentrations of 0.01 mg/m³; (**g**) Multiple scattering signals in the Raman scattering under chlorophyll concentrations of 0.1 mg/m³; (**h**) Multiple scattering signals in the Raman scattering under chlorophyll concentrations of 1 mg/m³.

### 4.3. Effect of SPF

In this study, the LUT method is used to simulate angle scattering under different SPFs in the MC program. Because fluorescence and Raman scattering have their own scattering phase functions, only the effect on the HSRL signal is simulated in this study. In Figure 12, we analyzed the effect of different SPFs. The lidar simulation results of HG (g = 0.924), HG (g = 0.81), ModHG, TTHG and FF SPF are compared in Figure 12a. The parameter settings of the lidar system are the same as those in Section 3.1, and the chlorophyll concentration is set as 0.1 mg/m³. The parameter g for ModHG and TTHG is 0.924. Figure 12a shows the color curves of these SPFs at different scattering angles. The Petzold curve is the seawater scattering phase function measured by early researchers [55], and the widely used HG SPF does not perform well, not matching the measured Petzold curve accurately for either small or large scattering angles. Because forward scattering is more likely to occur in seawater, the phase function has large peaks at small angles. In contrast, the FF SPF value is more consistent with the Petzold value. The TTHG SPF fits the Petzold values best when the scattering angle ranges from 20° to 120°.

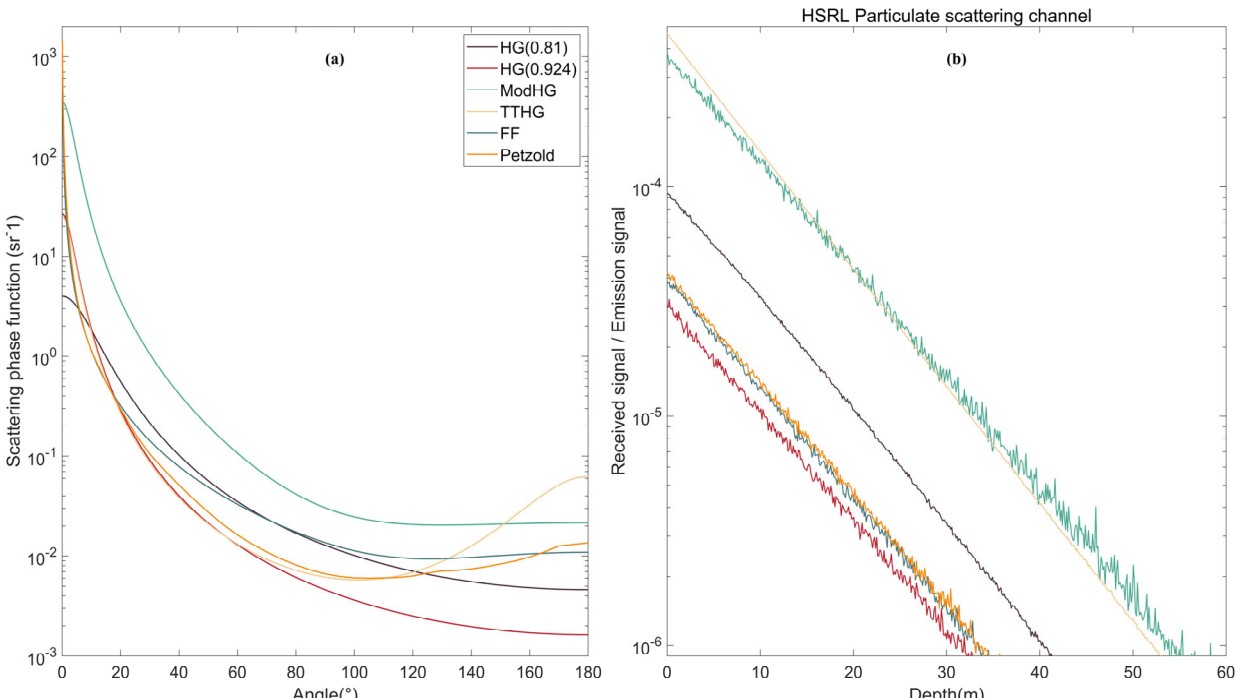

**Figure 12.** Effect of different SPFs on the HSRL echo signal: (**a**) Curves of SPFs depend on scattering angles; and (**b**) Simulation results using different SPFs.

Figure 12b shows the simulation results for different SPFs. The results of FF SPF and Petzold are relatively consistent because their SPF curves agree. The simulated value of FF SPF is marginally larger in magnitude than that of HG (g = 0.924) because the backscattering part of HG (g = 0.924) is lower than that of FF, and the probability of photon backscattering is lower. The signal intensity simulated by the HG (g = 0.81) is higher than that of the HG (g = 0.924) because the backscattering probability increases as the asymmetry Factor g decreases. ModHG SPF is generally high, which makes the simulation signal stronger but with more noise. The TTHG SPF is the largest in the backscattering part of all SPFS, photons are more likely to backscatter, and the simulation signal is stronger and less noisy. Thus, the appropriate SPF should be selected according to the real oceanic environment in lidar simulations, which is more accurate than simply using the HG SPF.

*4.4. Effect of Receiver FWHM*

As described in Section 2, fluorescence and Raman scattering have their own wavelength distribution functions. Therefore, different telescopes receiving spectral bandwidth (FWHM) will affect the intensity of the echo signal. In this study, we use a more accurate model for the $h_C(\lambda)$ function to study the effect of the receiver FWHM on the fluorescence echo to more truly reduce the fluorescence wavelength distribution function [33], as shown in Figure 13a. This model is the weighted sum of two Gaussian functions, one centered at 685 nm and the other at 730 nm:

$$h_f(\lambda) = P\frac{1}{\sqrt{2\pi}\sigma_{f1}}exp\left[-\frac{1}{2}\left(\frac{\lambda-685}{\sigma_{f1}}\right)^2\right] + (1-P)\frac{1}{\sqrt{2\pi}\sigma_{f2}}exp\left[-\frac{1}{2}\left(\frac{\lambda-730}{\sigma_{f2}}\right)^2\right] \quad (32)$$

where $P$ and $1-P$ are the weights of two Gaussian functions. We thus set $P = 0.75$, $\sigma_{f1} = 12.75$, $\sigma_{f1} = 25.50$ [30].

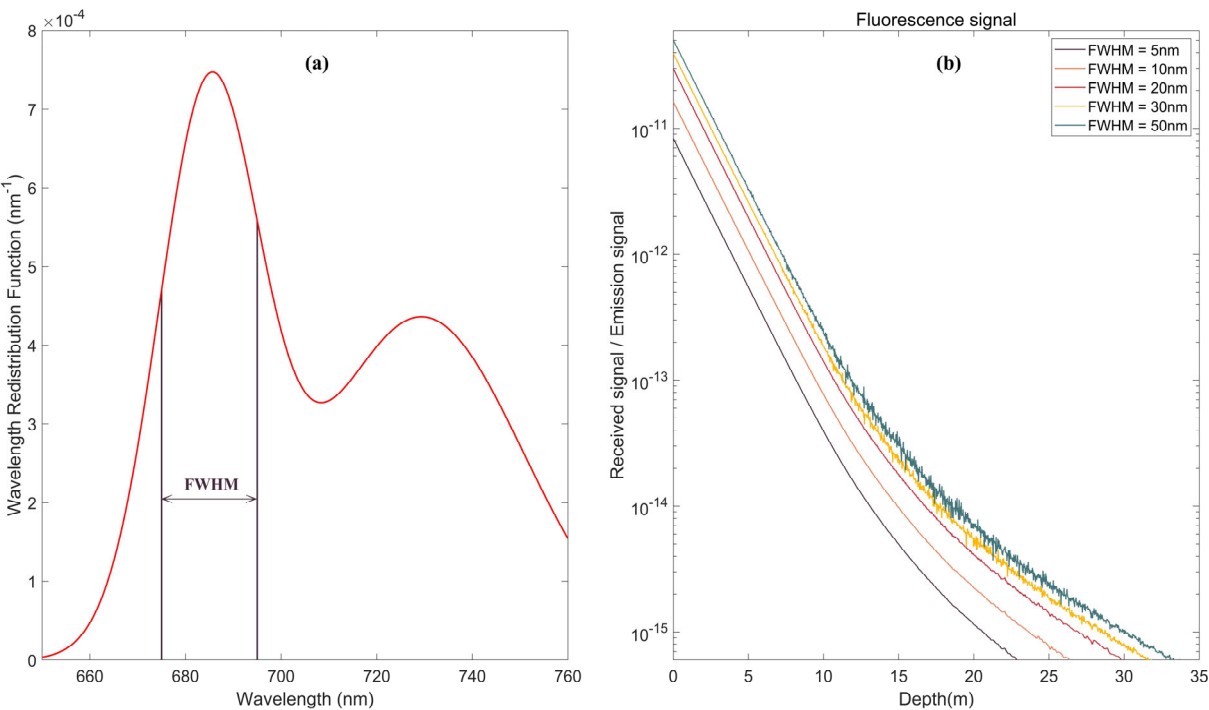

**Figure 13.** Effect of different FWHM on the fluorescence echo signal: (**a**) Wavelength redistribution function of fluorescence; and (**b**) Simulation results under different FWHMs.

The telescope primarily receives the signal of the 685 nm excitation wave crest; thus, the central wavelength of the FWHM is 685 nm. Fluorescence echo signals under different FWHMs are shown in Figure 13b. The intensity of the echo signal increases with increasing FWHM but also makes the telescope receive more background stray light. When the FWHM is 50 nm, the echo noise is markedly larger than that at 5 nm and 10 nm. As shown in the simulation results when the FWHM is 50 nm, the signal amplitude does not increase significantly, but the background noise increases sharply. Therefore, we set the FWHM of the receiving telescope between 20–30 nm to consider signal strength and noise control.

The emission wavelength distribution curve of Raman scattering is shown in Figure 14a. For an incident laser at 532 nm, the central wavelength of the emission function is approximately 650 nm. As shown in Figure 14b, because the emission spectrum width of Raman is narrower than that of fluorescence, the FWHM of the telescope set at 20 nm can receive more than 87% of the energy, and the noise is relatively small. After the FWHM is larger than 30 nm, the received signal amplitude is nearly unchanged. The curves at 30 nm and

50 nm basically overlap. Therefore, setting the FWHM of the receiving telescope to 20 nm can balance the signal intensity of Raman scattering and noise control.

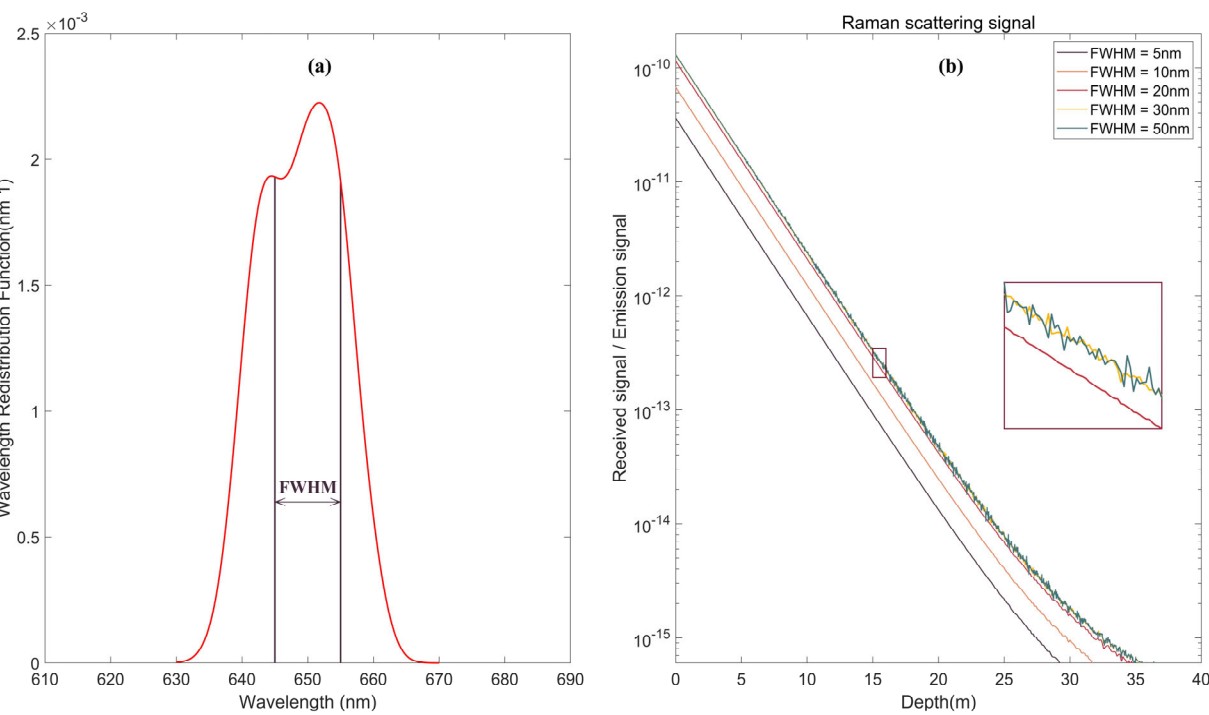

**Figure 14.** Effect of different FWHMs on Raman scattering echo signals: (**a**) Wavelength redistribution function of Raman scattering; and (**b**) Simulation results under different FWHMs.

### 4.5. Effect of Inhomogeneous Water

In this article, the chlorophyll-Gauss profile is used to simulate inhomogeneous water to analyze the effect of inhomogeneous water on lidar simulation [61]. The maximum layer depth of chlorophyll-a usually corresponds to the thermocline depth [62]. The optical properties of seawater at different depths are affected by the vertical structure of the chlorophyll-a concentration [4]; thus, the absorption and scattering coefficients of each layer must be calculated in the simulation. Thus, we can obtain a more accurate lidar echo signal. The vertical Gaussian distribution of chlorophyll $Chl(z)$ can be expressed as:

$$Chl(z) = Chl_1 + Chl_0 \cdot exp\left(-\frac{(z - z_{max})}{2\sigma^2}\right) \left(mg/m^3\right) \tag{33}$$

where $Chl_0$ is the sea surface chlorophyll concentration; $Chl_1$ is the background chlorophyll concentration; $z_{max}$ is the depth of maximum chlorophyll concentration; and $\sigma$ is the full width at half maximum chlorophyll value.

Figure 15a shows the Gauss profile of chlorophyll in inhomogeneous water under the conditions of $Chl_0 = 0.053$ mg/m$^3$ , $Chl_1 = 0.002$ mg/m$^3$ , $\sigma = 6$ m and $z_{max} = 25$ m . The equation in Section 2.2 is used to calculate the vertical profile of seawater optical properties for echo signal simulation. Figure 15b,c shows the simulation results of inhomogeneous water and homogeneous water, respectively. The chlorophyll concentration of homogeneous water is constant at 0.018 mg/m$^3$, and the simulated echo signal shows linear attenuation on logarithmic coordinates. This result shows that the intensity of the lidar signal decreases exponentially with increasing depth, following Beer's law. For inhomogeneous water, the HSRL particulate scattering echo signal shows a bulge corresponding to the depth of the chlorophyll profile bulge (Figure 15b). Based on this feature, the protrusions measured by lidar can be used to detect the subsurface chlorophyll maximum layer (SCML). The scattering rate of HSRL water molecular scattering itself is not affected by the chlorophyll

concentration, but due to the change in the attenuation rate and the effect of multiple scattering, the echo signal is less than that of homogeneous water in the high chlorophyll concentration segment of inhomogeneous water (Figure 15c). The fluorescence signal in the two types of water is also markedly different, particularly when the depth increases (Figure 15d). In inhomogeneous water, the surge of chlorophyll concentration excites higher fluorescence intensity, which makes the echo signal larger than that in homogeneous water. Meanwhile, the effects of multiple scattering were also offset near the peak depth of chlorophyll, and the curve showed an approximate linear decline. Real oceans are often inhomogeneous, which should be considered in lidar simulations.

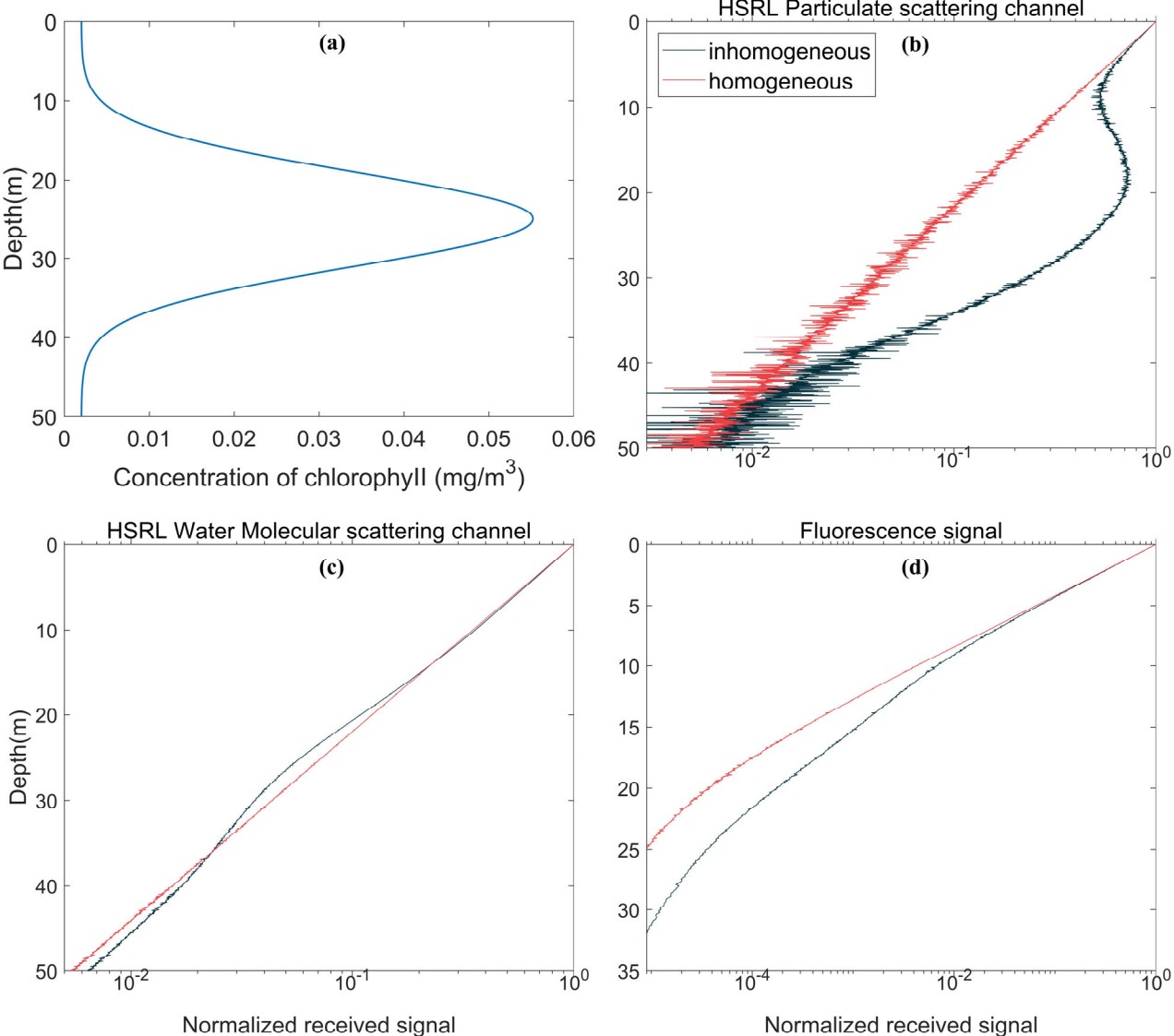

**Figure 15.** Effect of inhomogeneous water on lidar echo signal: (**a**) Vertical structure of the chlorophyll Gaussian profile; (**b**) Comparison of HSRL particulate scattering channel signals in homogeneous water and inhomogeneous water; (**c**) Comparison of HSRL water molecular scattering channel signals in homogeneous water and inhomogeneous water; and (**d**) Comparison of fluorescence signals in homogeneous water and inhomogeneous water.

## 5. Conclusions

We developed a novel tool to simulate inelastic scattering signals of oceanic lidar. The preliminary results showed that this tool can accurately describe a series of physical processes, such as light entering the ocean through the air–sea interface, scattering by water

molecules, scattering by underwater phytoplankton, generating stimulated fluorescence and stimulated Raman scattering and reflecting back to the receiver from the sea floor. By simulating various factors, we can draw the following conclusions:

(1) The higher the concentration of chlorophyll, the faster the speed of the HSRL echo signal decreases with depth. However, for fluorescence and Raman scattering signals, a high chlorophyll concentration can allow the receiver to detect deeper echo signals within its dynamic range. Under the same chlorophyll concentration, the fluorescence and Raman scattering simulated signals decay faster than the HSRL simulated signals.

(2) The simulation time is proportional to the chlorophyll concentration and indicates that turbid water produces more multiple scattering events and increases the multiple propagation paths. With increasing depth, the frequency of multiple scattering increases, and the intensity of the multiple scattering signal in each signal increases. For small FOVs, multiple scattering is so small that we can only consider single scattering; lidar attenuation is near that of water. For large FOVs, multiple scattering plays a major role in the total signal when the water depth increases to a certain extent.

(3) Different SPFs were used to assess their impact on HSRL particulate scattering signal modeling. The widely used HG SPF is not good for small or large scattering angles. The results of FF SPF and measured Petzold were relatively consistent. Therefore, in lidar simulations, an appropriate SPF should be selected according to the real oceanic environment.

(4) The effects of different FWHM receivers on fluorescence and Raman signals are simulated. The larger the FWHM, the higher the received signal intensity and the greater the background noise. Simulations show that a suitable FWHM of fluorescent lidar is between 20 and 30 nm, which is 20 nm for Raman lidar.

(5) For inhomogeneous seawater, the HSRL particulate scattering signal shows a bulge corresponding to the depth of the chlorophyll profile bulge. We can use this measurement feature to detect SCML. Inhomogeneous seawater also causes a change in the HSRL water molecular scattering signal and fluorescence signal. Thus, we should consider the influence of inhomogeneous water in oceanic lidar simulations.

Overall, the proposed multifactor simulation analysis of inelastic scattering shows that the effects of multiple scattering, SPF, receiver FWHM and inhomogeneous water must be considered, even though these are often ignored in existing lidar radiation transfer models. When the HSRL technology is more developed and can be applied to inelastic scattering signal detection, we plan to compare simulation data with measured data to verify the effectiveness of the proposed model. The relationship between the multiple scattering of oceanic lidar and many factors, such as field height, different types of water and observation geometry, requires further study. The wind-driven rough sea surface model will be incorporated into the simulation to study the influence of laser incidence angle and wind speed. Meanwhile, an underwater seabed Bidirectional Reflectance Distribution Function (BRDF) model will be established to study its effect on signals.

**Author Contributions:** Conceptualization, S.C. and D.P.; methodology, S.C. and P.C.; investigation, L.D.; writing—original draft preparation, S.C.; writing—review and editing, P.C.; funding acquisition, P.C. All authors have read and agreed to the published version of the manuscript.

**Funding:** This research was funded by the National Key Research and Development Program of China (2022YFB3901703), the Key Special Project for Introduced Talents Team of Southern Marine Science and Engineering Guangdong Laboratory (GML2019ZD0602), the National Natural Science Foundation (42276180; 41901305; 61991453), the Key Research and Development Program of Zhejiang Province (2020C03100), and the Donghai Laboratory Pre-research project (DH2022ZY0003).

**Institutional Review Board Statement:** Not applicable.

**Informed Consent Statement:** Not applicable.

**Data Availability Statement:** Not applicable.

**Acknowledgments:** The authors would like to thank the anonymous reviewers for their suggestions that markedly improved this article.

**Conflicts of Interest:** The authors declare no conflict of interest.

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
