# Peer review of "A New Semi-Analytical MC Model for Oceanic LIDAR Inelastic Signals"

_remotesensing, doi:10.3390/rs15030684_

Round 1

Reviewer 1 Report

In this paper, a Semi-Analytical Monte Carlo Simulation Model of oceanic LiDAR Inelastic Signals is established by integrating Fluorescence model and Raman scattering model into the Traditional Semi-analytical Monte Carlo Simulation Model, the echo signals of High Spectral Resolution LiDAR (HSRL) is simulated using this model. And the effects of chlorophyll concentration, multiple scattering, receiving field of view (FOV), scattering phase function (SPF), full width at half Maximum (FWHM) and inhomogeneous seawater on oceanic LiDAR echo signals are analyzed quantitatively.

Here are some questions after reviewing:

1. Whether the influences of the characteristics of the underwater seabed BRDF on the LiDAR echo signal are taken into consideration?

2. Why the influence of laser energy on the simulated signal of laser underwater echo was not discussed?

3. Why the influence of the laser incident angle on the simulated signal of the laser underwater echo was not discussed either?

4. The premise of the simulation model application made in the paper is the pelagic Class I water bodies, the water quality optical parameters of Class I water bodies are mainly calculated by the bio-optical model, in subsection 2.5 equation (19), the absorption coefficient of chlorophyll calculation formula is not detailed. Moreover, the water quality parameters have the greatest influence on the attenuation effect of the laser, near-shore water bodies and port water bodies are Class II water bodies. Can the effects of different types of water, especially Class II water, on the multiple scattering of laser transmission in water be discussed in more detail?

5. Lack of comparison experiments with the measured data, which needs to be improved in the subsequent study. Also note that the alignment of equation (29) should be consistent with other equations.

Author Response

We appreciate the time and effort that you have dedicated to providing valuable feedbacks on our manuscript. Those comments are all very valuable and helpful for revising and improving our article. We have studied the comments carefully and have made correction which we hope will be met with approval.

The manuscript (Ref. # remotesensing-2130995) has been carefully revised. For easy reading, all major changes are highlighted using the “Track Changes” function in the revised manuscript. Point-by-point answers to the comments and suggestions are given here. Please see the attachment.

Reviewer 2 Report

General comments

This manuscript is well written and has a lot of scientific merit. It presents a study of a new semi-analytical MC model for oceanic lidar inelastic signals, which is important for ocean exploration. Based on the optical properties of seawater, this model simulates various inelastic echo signals of oceanic lidar and analyzes the multiple scattering of particulate and molecules in detail. In particular, it takes into account chlorophyll concentration, FOV, SPF and FWHM in modeling, and can provide a reference to the design of HSRL lidar systems. This works is beneficial to recognize the mechanism of lidar remote sensing of ocean. The structure is logical, the figures are of good quality and the historical background has given credit as is appropriate. Overall, I find it suitable for publication in Remote Sensing, with minor revision regarding the following points.

Specific comments

L 93: This flowchart represents the fluorescence simulation, please check the title.

L 112: What is the n means in eq.3

L 181: What is the difference between the FWHMFLUOR and the receiver FWHM

L 314: mg/m3 → mg/m3

L 425: “Section III” or “Section 3”. Please describe in a uniform way.

L 435: “the higher the chlorophyll concentration is, the deeper the echo signal that can be received in the telescopic dynamic range” According to the figure, whether this conclusion is only for fluorescence and Raman signals. Please check it.

L 439: “three signals” or “four signals”? Please check it.

L 456: Is there parameter “g” for “ModHG” and “TTHG”? What are the set params of FF?

L 488-489: Add references.

L 537: Whether the principle of the effect of inhomogeneous water on fluorescence can be explained?

Author Response

(The authors gave the same response as above.)
